# The Moment Criterion of Anthropomorphicity of Prosthetic Feet as a Potential Predictor of Their Functionality for Transtibial Amputees

**DOI:** 10.3390/biomimetics8080572

**Published:** 2023-11-28

**Authors:** Mark Pitkin

**Affiliations:** 1Poly-Orth International, Sharon, MA 02067, USA; mpitkin@tuftsmedicalcenter.org; 2Department of Orthopaedics and Physical Medicine and Rehabilitation, Tufts University School of Medicine, Boston, MA 02111, USA

**Keywords:** limb prosthetics, design requirements, moment critetion of anthropomorphicity

## Abstract

The purpose of this paper is to discuss a new quantitative mechanical parameter of prosthetic feet called the Index of Anthropomorphicity (*IA*), which has the potential to be adopted as an objective predictor of their functionality. The objectives are to present the research findings supporting the introduction of *IA* and unify previous results into a coherent theory. The *IA* is founded on the moment criterion of the anthropomorphicity of prosthetic feet. The term “anthropomorphicity” is defined for this application. Studies with a small number of human subjects and prostheses have shown that the value of the parameter is positively correlated with patient comfort and with the restoration of certain normal gait characteristics. Confirmatory studies with controlled human trials and mechanical tests with a wider selection of prosthesis types can give prosthesis manufacturers a new criterion to follow in the design process, and prosthetists may use the *IA* for selecting more suitable prostheses for a patient’s comfort and health.

## 1. Introduction

### 1.1. Problem of Overloading the Residuum

Problems with the residuum’s overloading lead to discomfort, pain, and damage to the residuum skin in patients with traditional socket attachment and to the residuum bone in patients with direct skeletal attachment, or osseointegration. The skin within a socket, or the bone marrow canal in cases of osseointegration, must withstand the excessive compressive and frictional forces for which these tissues are poorly adapted [1,2,3,4,5].

The presence of severe trauma, vascular disease, diabetes, or malignancy makes the residuum skin even more vulnerable to tissue breakdown, inflammation, infection, and other malignancies [4,6]. As a consequence of pain, discomfort, and infection, amputees may reduce their prosthesis use and become more limited in their walking capacity, increasing the chance of partial or total prosthetic abandonment and deterioration in the quality of life [3,7,8]. 

Mitigating residuum skin complications with adjustable sockets and liners [9,10] does not fully address the problem of prostheses being fabricated in the absence of objective design criteria. While osseointegration replaces the socket attachment of the prosthesis by its direct attachment to the transdermal pin implanted into the residuum’s bone [11,12,13,14], the technology faces the same intrinsic challenges of bone–implant interfaces [15,16]. 

A root cause of the issues at the body–device interface is the high pressure consistently applied to the residuum with every step. To alleviate the pressure, patients develop compensatory gait strategies that place undue pressure on other areas of the musculoskeletal system, leading to arthritis in the uninvolved leg and other negative externalities. It is imperative to reduce the pressure on the residuum a priori, through improved prosthetic design and prescription, rather than compel the amputee to find unhealthy coping strategies.

It is desirable to predict how a person with transtibial amputation will perform with a given prosthesis before it is worn just by knowing certain mechanical characteristics of this device. However, this task is still not a matter of objective science, but of art and experience [17,18,19,20,21] and there are no quantifiable measures to guide prosthesis prescriptions beyond the prosthetist’s experience and the manufacturer’s recommendations [18,22,23,24,25]. Research to aid the clinician knowing which specific foot–ankle mechanisms to recommend is limited. 

Prosthetic feet and ankles are often studied and compared on the basis of how they substitute for physiologic foot functions. Perry [26], for example, lists the three main functions of the physiologic foot as shock absorption, weight-bearing stability, and progression. Valmassy [27] classifies the five functions of a leg prosthesis as load bearing, leverage, shock absorption, balance, and protection. In neither of these classifications, nor in any others, however, are these characteristics quantified.

### 1.2. Current Regulatory Status of Prosthetic Feet

The current regulatory status of prosthetic feet is as follows. The FDA categorizes all medical devices into one of three classes—Class I, II, or III—based on their risks and the regulatory controls necessary to provide a reasonable assurance of safety and effectiveness. Class I devices pose the lowest risk to the patient and Class III devices pose the highest risk.

Ankle and foot components are classified as Class I and are exempt from premarket notification procedures and from good manufacturing practice requirements (FDA, *21CFR890.3420*) [28]. The exemption is based on the recognition that the ankle and foot are *external limb prosthetic components*. To date, the only prosthetic safety parameter that is measured is the structural strength of the device itself, and it is regulated by ISO 22675:2016 [29], ISO 22523:2006 [30], ISO/TS 16955:2016(en) [31] and ISO 10328:2016(en) [32]—both developed by the ISO/TS 168, Prosthetics and Orthotics Committee [33].

ISO 22675 and ISO 10328 contain a warning that the standard “*is not suitable to serve as a guide for the selection of a specific ankle-foot device in the prescription of an individual lower limb prosthesis and any disregard of this warning can result in safety risk for amputees.” At the same time, it states that this standard “allows other applications directed to the assessment of specific performance characteristics of ankle-foot devices and foot units that may be of relevance in the future as “each sample of ankle-foot device or foot unit is… free to develop its individual performance under load”* [29]. Also, 16955:2016(en) describes the quantitative methods to assess key performance indicators without attempting to establish the correlation between these measures and the relevant measures of prosthetic user benefit.

The American Orthotic and Prosthetic Association (AOPA) conducted a double-blind, randomized study to evaluate patient preferences for maximal sagittal moment during dorsiflexion, and found that patients prefer the *compliant* category of prosthetic feet compared to the *intermediate* and *stiff* categories [24,25]. The progress of dorsiflexion was evaluated versus the percent of the stance time duration, and a definition of the compliance of the prostheses (*compliant, intermediate,* and *stiff*) was linked to the maximal value of the prosthetic ankle’s moment generated at the end of the dorsiflexion period of the stance phase of gait. As with the ISO 22675:2016, the study report begins with the warning: “*At no time during this project did AOPA ever intend to develop clinical standards of care for these feet. The project’s findings relate only to appropriate coding and do not speak to what foot is most clinically appropriate for a particular patient*”.

### 1.3. Motivation for the Current Study

Anthropomorphicity in this branch of research refers to the closeness of the moment–angle dependencies in prosthetic and in anatomical ankles. The motivation for the current study was to introduce a moment criterion of anthropomorphicity as a numerical criterion for designing and further selecting prosthetic feet. The objective of implementing such a criterion is to mitigate the overloading of the residuum by developing prosthetic foot and ankle systems reproducing key features of the anatomical prototype over the *entirety* of the stance phase. A specific feature, namely the moment–angle dependence, may be verified in the mechanical bench tests of the prostheses and in the biomechanical trials of patients wearing various prostheses. The dependence obtained from biomechanical trials for the anatomical ankle joint may serve as the desired target to be replicated in the prosthetic ankle and called the *moment criterion of its anthropomorphicity.* For the objectivity of evaluating the level of anthropomorphicity, a numerical characterization of the shape of the moment–angle curves is required. 

This paper presents such a characterization, called the Index of Anthropomorphicity (*IA*), and demonstrates its potential to be a predictor of the preference and performance of patients before they begin using the prosthesis. This methodology has been introduced by the author and further developed jointly with his colleagues over a long period of time [34,35,36,37,38,39,40,41]. Therefore, it becomes necessary now to unify the hypotheses and outcomes of previous and recent studies into a coherent theory as a first step toward developing an objective standard of the functionality of prosthetic feet.

## 2. Materials and Methods

In this section, we consider stages and rationales in the development of the Index of Anthropomorphicity and the verification of its correlation with performance and perception by persons with transtibial amputation.

### 2.1. Normal Ballistic Synergy and Compensatory Prosthetic Synergy

#### 2.1.1. Ballistic Synergy in the Norm

Ballistic synergy in gait—the coordinated combination of free and limited rotations in the ankle, knee and hip joints utilizing optimal relationships between the ground reactions and the force of gravity, allowing muscles to work in an economical and eccentric mode—is an indicator of the soundness of the musculoskeletal apparatus and of the neuromuscular control system [41]. In the norm, when the vertical loading on the leg reaches its maximum (at about 40% of the stance phase), dorsiflexion proceeds with low resistance. It then quickly increases nonlinearly, until ankle angulation is locked and the heel is lifted by inertia, resulting in a concave pattern in the moment–angle diagram [41]. This steep increase in the ankle resistance to further dorsiflexion is referred to as deceleration during dorsiflexion [42], and is a means to slow down the movement of the body’s center of mass and facilitate heel-lift. Knee flexion/extension during the stance phase is known as the “third determinant of normal gait” of six such determinants [43]. Its purpose is to absorb shock after the heel strike and decrease energy consumption by lowering the maximum elevation of the center of gravity of the body in mid-stance. The mostly concave shape of the curve in the dorsiflexion period of the stance indicates that the initiation of dorsiflexion does not encounter a large amount of resistance from the plantarflexors. The maximum magnitude for the moment around the talocrural joint (articulation of the ankle in the sagittal plane) averages from 80 to 120 Nm. A similar concave nonlinear pattern is seen in the talocalcaneal joint (frontal articulation) with a maximum moment of resistance of 23–25 Nm [44]. 

One can immediately sense the negative effects of increased resistance to dorsiflexion in the ankle by trying to walk in ski boots where, due to the ski boot’s stiff ankle zone, the resistive moment from the boot’s cuff to the shin increases and creates the sensation of “shin-bang” (Figure 1). The experience of an amputee walking in a prosthesis with excessive resistance to angulation—that is, with a convex moment–angle dependence—is, over the long run, analogous.

#### 2.1.2. Compensatory Synergy in Prosthetic Gait

The moment–angle dependency of the prosthetic ankle and foot is critical because an individual with amputation must create a bending moment by the residuum to overcome the resistance to flexion–dorsiflexion in the prosthetic ankle. To achieve it, the amputee generates with the residuum and applies to the socket a pair of equal and opposite forces *F*, *−F* (Figure 2a) [34]. When a prosthetic ankle is excessively resistive to angulation, a larger bending moment is required from the residuum, and a correspondingly greater loading is applied back to it. This is illustrated with the convex graph in Figure 2, a-1. Conversely, if the ankle moment–angle graph is concave, as shown in Figure 2, b-2, allowing for almost free initial dorsiflexion, the force couple *F*_1_, *−F*_1_ (Figure 2b) and corresponding loads on the residuum are lower [34]. Excessive cyclic pressures on the skin and soft tissues from the socket are painful and can be harmful, and the amputee consciously or intuitively develops strategies to avoid them. 

As demonstrated in [45], the pain-protective strategy is most prevalent when the tibia passes the vertical position and the vertical component of the ground reaction force reaches its greatest value (at about 40% of the stance period of gait). That is the time when, to continue dorsiflexion without excessive and therefore painful loads from the socket walls, the residuum has to meet the lowest possible resistance from the ankle unit of the prosthesis. As a pain-protective means, transtibial amputees try to avoid bending the rigid ankle unit, and that synergistically decreases the maximal knee stance-flexion angle in the involved leg despite the fact that the anatomical knee joint of that leg is intact. 

The average ROM in the transtibial amputee’s knee joint of the involved leg is approximately half that of biological joints (7 versus 15 degrees, respectively) [46]. As there is no anatomical basis for such reduced flexion, it is the exceedingly rigid foot and ankle which is responsible for this. Indeed, the amputee intuitively decreases the ROM in the knee to avoid pain in the residuum, as illustrated in Figure 2. With an asymmetrically lower loading of the involved leg, the amputee automatically imposes greater loading on the uninvolved intact leg. This is the main reason why transtibial amputees have an increased risk of developing osteoarthritis in the knee and hip of their intact limb [47,48]. Overall, increased asymmetry in the maximal knee stance-flexion angle, followed by the synergistically increased asymmetry in other gait parameters, is an objective indication of the pain-protective strategy used by amputees [41].

### 2.2. Anthropomorphicity of Prosthetic Feet and Evolution of Its Criteria

The term anthropomorphism (*Greek ánthrōpos* (*human*) + *morphē* (*shape* or *form*) refers to the attribution of human characteristics to nonhuman phenomena and objects, but there is no common convention about its meaning when applied to specific situations. Nevertheless, the entire history of prosthetics is driven by a desire for anthropomorphicity, either conscious or intuitive, in selecting the anatomical leg’s features to be mimicked in the artificial limb [41]. The traditional criterion of anthropomorphicity is “structural”, related to the need to compensate for the leg’s shortening after amputation. Another criterion is “cosmesis” when visual, tactile, and other anatomical characteristics are to be met. Other criteria of anthropomorphicity have come from the biomechanics of locomotion in the norm and with prostheses [41]. 

In 1975, the author formulated the requirement of prosthetic feet to be *anthropomorphic* in the sense of the device imitating the concave moment–ankle dependency generated by the anatomical ankle joint during gait [39], an idea since adopted by various research groups [49,50,51,52,53,54,55,56,57,58,59,60,61,62]. He developed a mathematical model of the mobility of the human foot in the norm [39], which explained the classical biomechanical result of the nonlinear increase in resistance to the spreading of the foot under a vertical load from the tibia (Wright and Rennels, 1964) [63]. 

The concave pattern of the resistive moment in an anatomical-like compliant joint can be generated by different means; however, to be closer to the anatomical prototype, a cam-rolling structure was selected, in which contact surfaces roll when there are any changes in the relative positions of either component. Thus, a prosthetic foot mechanism was synthesized with a relative rolling of its parts, resulting in the commercially available Rolling Joint Foot and Ankle prosthesis (RJFA), WillowWood Global (formerly known as Ohio Willow Wood, Co), Mt. Sterling, OH 43143, USA [34,45,64,65,66]. The RJFA prosthesis was further developed and marketed by WillowWood Global (formerly known as Ohio Willow Wood Co.), Mt. Sterling, OH 43143, USA, under the Free-Flow Foot and Ankle (FFF) name [64] (Figure 3). 

In mechanical tests, the FFF exhibited the concave moment–angle dependency observed in the norm, in contrast with the convex shape produced by the SACH foot, United prosthetics, Inc. 295 Columbia Rd., Dorchester, MA, 02121, USA and Flex Foot, Irvine, CA 92618, USA [67] (Figure 4). In biomechanical tests, as predicted, the FFF generated lower forces and pressures on the amputee’s residuum than other prosthetic feet that were tested, and it was associated with a more positive perception of the prosthesis [36,45,66]. That reduction in forces and pressures on the residuum was achieved due to the lower moment of resistance in the middle of the stance when the ground reactions reach maximal magnitude, which is seen through the concavity of the moment–ankle diagram. This allowed participants to restore normal values in the maximal knee stance-flexion angle (a statistically significant difference with 95% confidence was found [67]) (Figure 5), leading to increased comfort and decreased pain [45,65,66,67]. 

#### 2.2.1. Index of Anthropomorphicity and Its Relation to Bending Moments

The Index of Anthropomorphicity (*IA*) is a quantitative, automatically computable measure of the concavity/convexity of the moment–angle relationship, relying on a technique first described in 2000 [36] (Figure 6a) and then replicated in [62,68]. 

Let K1 (Figure 6a) be a measure of ankle stiffness derived from the first portion of the stance phase; let K2 be the corresponding measure of stiffness in the second stance portion. *IA* is defined as K2-K1. In normal gait, K2 is greater than K1, like in any other concave curve. Accordingly, *IA* is positive in normal gait and in prosthetic gait with a concave moment–angle graph. It is negative in prosthetic gait with a convex moment–angle graph (Figure 6b).

#### 2.2.2. Automated Procedure to Compute the Index of Anthropomorphicity

An automated technique for extracting the stiffnesses K2 and K1 to compute *IA* was described in [40]. The moment–angle curve is first characterized by the duration between relevant events occurring during the flat foot phase of the support period corresponding to the beginning and end of dorsiflexion, and the point of curvature where the second derivative of the moment–angle curve is identified as zero. The slopes K2 and K1 of the regression line of the moment–angle curve before and after the point of curvature are calculated, so that a positive *IA* (K1 < K2) or negative *IA* (K1 > K2) corresponds to a concave or convex shape, respectively. 

An observational case–control biomechanical study with three below-knee amputees with osseointegration and four able-bodied subjects as a control was conducted [38]. Amputees were walking with their own prosthetic ankle/foot units: RUSH (Proteor, Tempe, AZ85282, USA), Trias (Ottobock, Salt Lake Citi, UT 84120, USA), Triton (Ottobock, Salt Lake Citi, UT 84120) and with the Free-Flow Foot (FFF). Moments were recorded wirelessly with a portable kinetic system (iPecs Lab, RTC, Dexter, MI, USA), and the ankle angle was extracted from the videos (Canon, IXUS, US). 

The stiffness characteristics of the sound ankle were extracted for four able-bodied participants (2 males, 2 females, 25 ± 2.5 years, 1.71 ± 0.12 m, 68 ± 1.72 kg). Participants were recruited by New England Sinai Hospital between January and April, 2014. Able-bodied participants walked 3 trials at a self-selected speed in the gait lab equipped with 6 cameras (Vicon Motion Analysis System, Oxford, UK) and 2 Kistler force plates (Kistler Instrument Corp., Novi, MI, USA). Dorsiflexion angle data were extracted from the 3D motion capture with the video and force plate sampling frequency of 200 Hz and using the Modified Helen Hayes full-body reflective 9 mm marker set. Bending moment data were calculated using inverse dynamics.

#### 2.2.3. Outcomes of the Observational Study

Values of K1, K2, and Indices of Anthropomorphicity (*IA*) for subjects wearing their own prosthetic feet, and then the Free-flow foot, and in able-bodied subjects, are presented in Table 1**.** The relationships between the Indices of Anthropomorphicity and the bending moments generated by the sound and prosthetic ankles are shown in Figure 7. The means ± standard deviations of the Indices of Anthropomorphicity are shown on the x axis (see Table 1). The values of bending moments coincide with the maximal values of the vertical ground reactions (approximately 40% of the stance period), when the loads on the residuum are the greatest [38]. 

Prosthetic feet can be quite clearly categorized as anthropomorphic or not. The RUSH, TRIAS, and Triton feet (owned by the subjects) which generated convex moment–angle curves all had negative *IA*s: IA=−2.97 ± 2.37. The Free-Flow Foot (FFF) had a positive, anthropomorphic *IA* = 2.68 ± 1.09, although its magnitude was less than in the norm (*IA =* 5.88 ± 0.93).

As predicted, the anthropomorphic FFF foot prosthesis reduced the maximal bending moment applied to the implant compared to the three non-anthropomorphic feet worn by the subjects, with a significant 25% reduction. 

## 3. Discussion

More than 2.4 million American patients were estimated to have limb loss in 2021, and the number is expected to rise to 3.6 million in the upcoming decades [69,70]. The incidence of lower limb amputation will rise concurrently [71,72]. The lifetime healthcare costs for an individual living with limb loss exceed USD 500,000 and that number consistently increases [73,74,75].

Outcomes of rehabilitation are impacted by impaired mobility, pain, and the pathological overloading of the remaining limb [76,77,78]. In the course of prosthetic rehabilitation, special attention has to be paid to residuum skin conditions in the socket, or the bone marrow canal in case of osseointegration. Due to pain, discomfort, infection, or loosening, amputees may reduce their prosthesis use, increasing the chance of prosthetic abandonment and deterioration in the quality of life [3]. Nowadays, the common understanding prevails that prosthetic feet with properly selected moments of resistance to angulation (stiffness) in the sagittal and frontal planes and in axial rotation might improve stability, which will in return provide enhanced safety [21,60,61,62,79,80,81,82]. However, characterizations of the moment–angle diagram differ between researchers.

The approach by the author has been that the shape of the moment–angle diagram has to meet the moment criterion of anthropomorphicity with a specific numerical parameter of the dependency, which is associated with the restoration of normal ballistic synergy in gait [41]. As such, it can be included in the manufacturer’s specification of each prosthetic foot to help the rehabilitation team maximize the patient’s satisfaction and performance. The Index of Anthropomorphicity discussed in this paper serves this purpose. This parameter is a quantification of the moment criterion of anthropomorphicity in the sense of the device imitating the concave moment–ankle dependency generated by the anatomical ankle joint during gait.

We spoke about the moment criterion of anthropomorphicity in relation to the nonlinear moment–angle dependency in the ankle joint. At the same time, this criterion is applicable to the knee, hip, and any other articulating joint. The sharp increase in resistance to angulation is a protective feature, protecting the anatomical joint from damage when angulation comes closer to the border of its range of motion. It can also be considered as a “charge-release” mechanism of energy transformation during locomotion. Therefore, the moment criterion merits its implementation in limb prosthetics in general.

The primary limitations of this study are the low number of prosthetic feet tested for which the Index of Anthropomorphicity has been calculated, and the low number of amputees and able-bodied subjects. Future studies should address these small-sample limitations. It will be important to conduct follow-up comparative studies in the longevity of osseointegrated attachment in patients using feet with and without anthropomorphic *IA*. That would validate the positive effect of a decrease in loading the abutment, which otherwise could not be noticed by the amputee as readily as with the socket attachment. Future studies should also involve the calculation of the Index of Anthropomorphicity from the gait analysis of various categories of able-bodied persons (stratified by age, sex, physical activity, etc.). In addition to traditional motion laboratory techniques with multiple video cameras and force plates, such data could be obtained outside the laboratory in the patient’s home using the wavelet transform of signals obtained from inertial ProMove mini sensors [83] or other portable systems [84,85].

## 4. Conclusions

A quantitative anthropomorphic criterion—the Index of Anthropomorphicity (*IA*)—for comparing prosthetic feet has been proposed, based on the closeness of the moment–angle dependency between the anatomical ankle and prosthetic ankles with a high *IA*. A preliminary study showed that a prosthetic foot with an anthropomorphic moment of resistance to angulation in ankle tends to decrease the maximal loads applied to the residuum. Indices were computed programmatically for able-bodied subjects and amputees, and an *IA =* 5.88 ± 0.93, computed in able-bodied subjects, can be considered as the baseline value for comparisons with *IA* of prosthetic ankle joints. A prosthetic foot and ankle designed with the moment criterion of anthropomorphicity showed an *IA* closest to that observed in the norm.

Obtaining the Index of Anthropomorphicity in mechanical testing and its inclusion into specifications may be used for selecting prosthetic feet in patients both with socket attachment and with osseointegration. In the future, the approach and technology developed for the quantitative comparison of design mechanical outcomes of prosthetic feet should be applied to a larger group of participants and a more representative group of prostheses, as proposed in this study. With the proper validation of the calculated *IA* in widely prescribed prosthetic feet and the investigation of how this new quantitative parameter correlates with overall functionality and comfort of prostheses, the Index of Anthropomorphicity of a prosthetic foot identified via mechanical tests could be a predictor of an amputee’s performance with that prosthesis. This approach would guide the designer of prosthetic feet and facilitate the selection of the prosthetic foot by decreasing the number of trials needed with different prostheses.

## Figures and Tables

**Figure 1 biomimetics-08-00572-f001:**
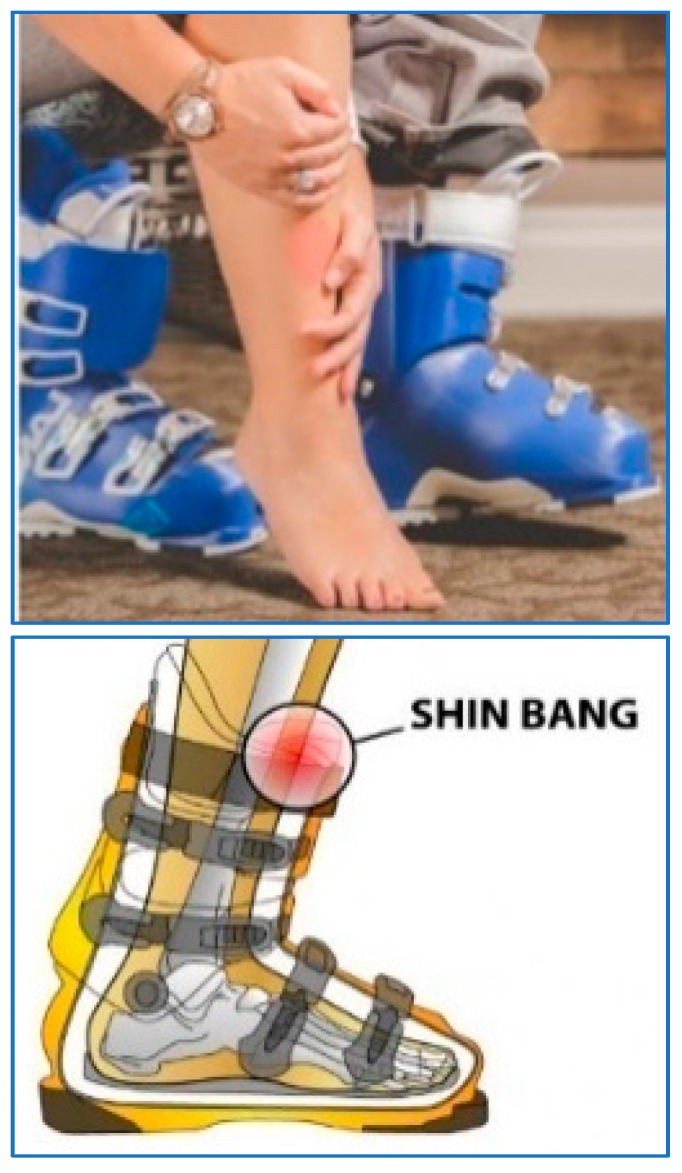
Shin-bang due to stiff ankle zone in the boot.

**Figure 2 biomimetics-08-00572-f002:**
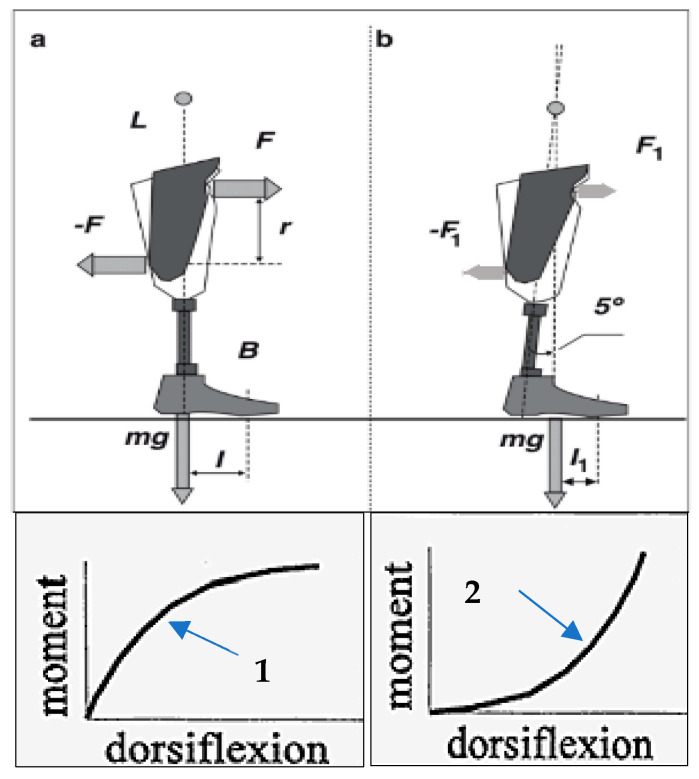
Convex (1) and concave (2) moments of resistance to dorsiflexion in a prosthetic ankle affect the forces applied to the amputee’s residuum— schematics in (**a**) and (**b**) illustrate that 5^0^ of free mobility in the middle of stance phase produces concave moment of resistance resulting in significant decrease in the load from the socket to the residuum (*F***_1_**) compared to load *F* generated by the socket when the ankle is solid [34], 1995.

**Figure 3 biomimetics-08-00572-f003:**
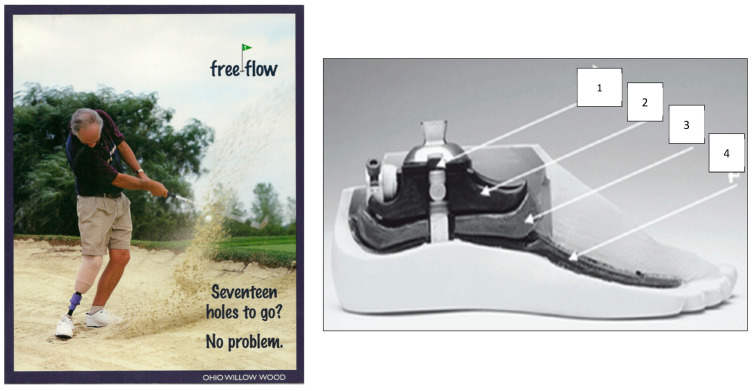
**Left**: Rolling Joint “Free-Flow Foot and Ankle” by WillowWood Global (formerly known as Ohio Willow Wood Co.) **Right**: 1—tuning screw for adjustment of initial stiffness; 2—tibial component able to roll along the elastic cushion 3; 4—base talar component [64].

**Figure 4 biomimetics-08-00572-f004:**
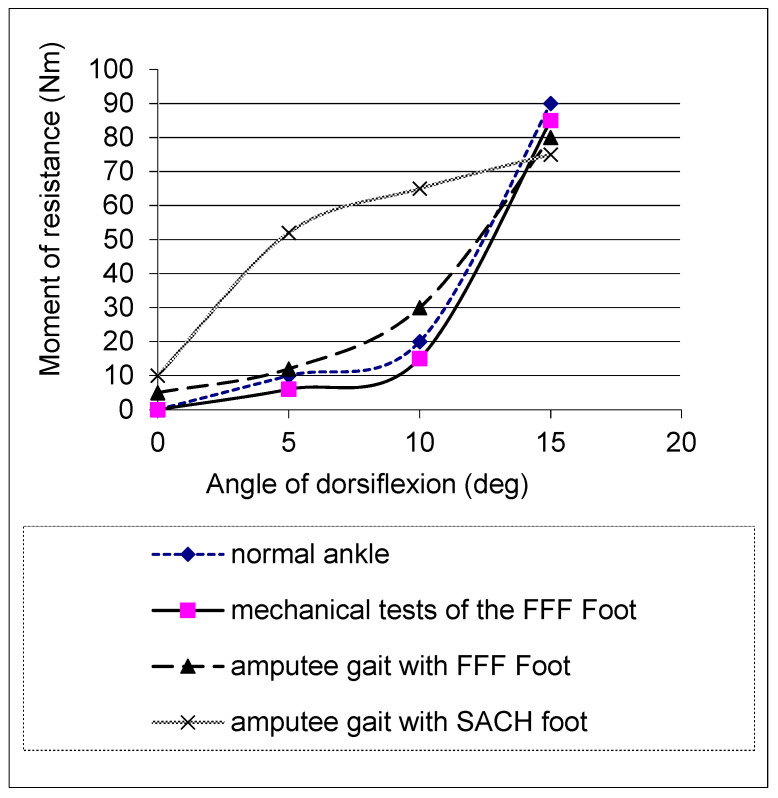
Moment of resistance in normal and prosthetic ankles vs. dorsiflexion angle [67].

**Figure 5 biomimetics-08-00572-f005:**
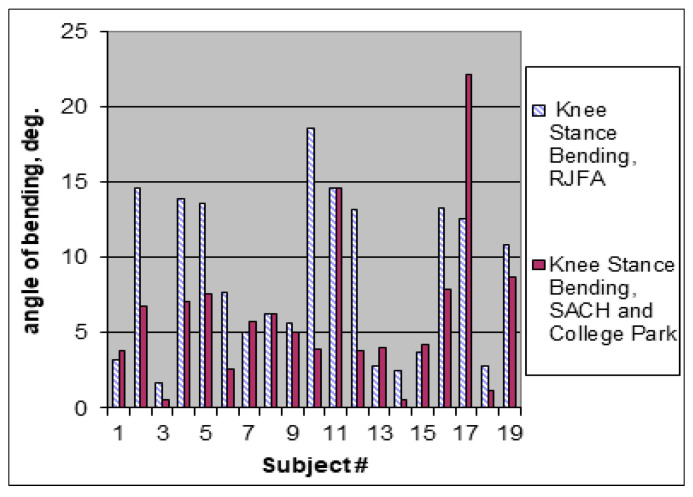
Influence of prosthetic foot on the ability to bend the knee during the stance period of gait [67].

**Figure 6 biomimetics-08-00572-f006:**
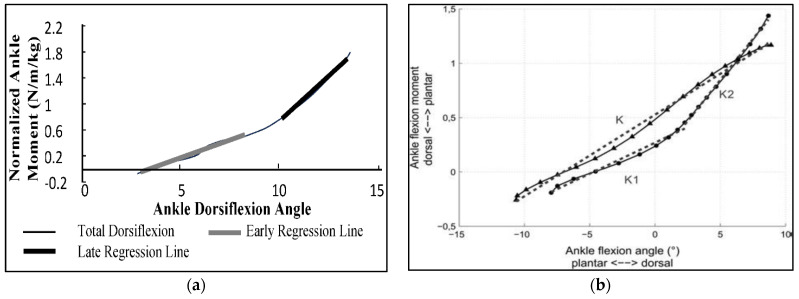
(**a**) Normalized dorsiflexion ankle moment vs. angle in norm. Slopes of shaded regression lines represent joint stiffness during early and later portions of stance [36], 2000. (**b**) Ankle moment vs. angle with computed stiffness (grey dashed lines); level walking in norm (black disc), K1, and K2) and transfemoral amputees (black triangles, K) [62], 2014. Reprinted with permission from Taylor & Francis, License #5635101073664. Ankle stiffness was computed through a linear interpolation of the relationship between the angle and the moment.

**Figure 7 biomimetics-08-00572-f007:**
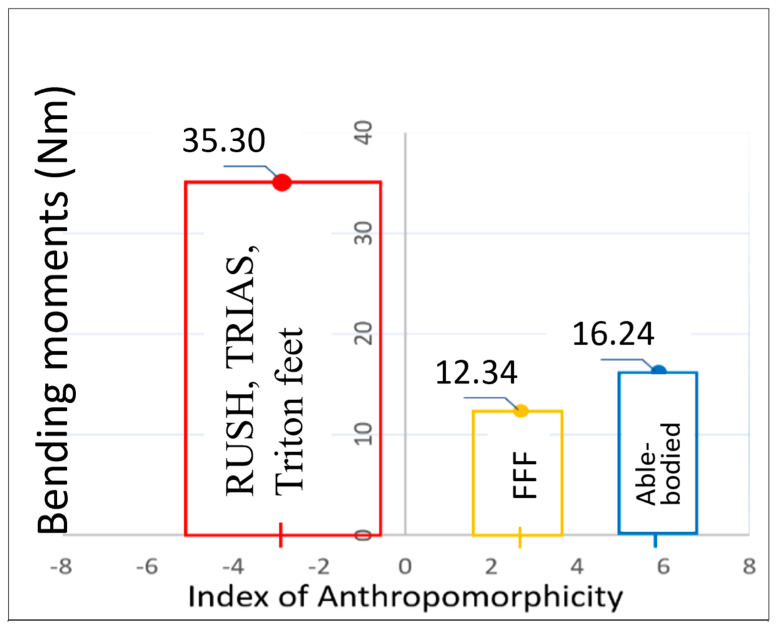
Bending moments in able-bodied ankles (blue); FFF (orange); amputees’ own prosthetic feet (red) vs. mean ± SD of the Index of Anthropomorphicity (*IA*).

**Table 1 biomimetics-08-00572-t001:** Values of the K1, K2, and *IA*s in subjects walking with their own prosthetic feet, then with Free-flow foot, and in able-bodied subjects.

	Gait with the Subjects’ Own Prosthetic Feet	Gait of the Subjects’ with Free-Flow Foot	Able-Bodied Subjects
K1	5.299 ± 1.682	1.001 ± 0.392	3.053 ± 2.053
K2	2.333 ± 1.585	5.299 ± 1.682	8.936 ± 1.804
*IA*	−2.966 ± 2.369	3.682 ± 0.912	5.883 ± 0.929

## Data Availability

The data presented in this study are available in the cited articles: [34,35,36,37,38,39,40,41,45,64,65,66,67].

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
