# Peer review of "The Moment Criterion of Anthropomorphicity of Prosthetic Feet as a Potential Predictor of Their Functionality for Transtibial Amputees"

_biomimetics, 2023, doi:10.3390/biomimetics8080572_

Round 1
Reviewer 1 Report (Previous Reviewer 5)
Comments and Suggestions for Authors
biomimetics-2744510-peer-review-v1
The revised and resubmitted manuscript, which focuses on prosthetics, offers a unique perspective on the application of an anthropomorphicity index and its correlation with prosthetic function and adaptability. This information is valuable in this field.
However, I strongly recommend professional English proofreading to enhance the readability and flow. In particular, it is beneficial to avoid paragraphs composed of overly short sentences.
Comments on the Quality of English LanguagePlease check my comments
Author Response
Please see the attachment

Reviewer 2 Report (Previous Reviewer 4)
Comments and Suggestions for Authors
In this paper, the author discusses a new quantitative mechanical parameter of prosthetic feet, called the Index of Anthropomorphicity (IA), which has the potential to be adopted as an objective predictor of their functionality.
I read the resubmitted manuscript with attention and I have only one comment. All figures should be modified and enlarged because they are low quality! In this matter, the article requires correction before publication.
Author Response
Please see the attachment

This manuscript is a resubmission of an earlier submission. The following is a list of the peer review reports and author responses from that submission.
Round 1
Reviewer 1 Report
Comments and Suggestions for Authors
Dear author, thank you for addressing my concerns and comments.
Reviewer 2 Report
Comments and Suggestions for Authors
Dear Author,
thank you for resubmitting the article. You done a great work. Congrats
Reviewer 3 Report
Comments and Suggestions for Authors
Authors improved the manuscript according to Reviewers' requests
Comments on the Quality of English LanguageAuthors improved the manuscript according to Reviewers' requests
Reviewer 4 Report
Comments and Suggestions for Authors
In this paper, the author discusses a new quantitative mechanical parameter of prosthetic feet, called the Index of Anthropomorphicity (IA), which has the potential to be adopted as an objective predictor of their functionality.
I read the article with attention and I have several comments, which I've shared below.
First, the article is presented by one author, but its text is addressed to a few authors. There is written "We” instead of "I" many times and referred to common research [32-40] with Quesada, Colvin, Frossard, and Leech. Why were they omitted from this article?
Also, I noticed that the title of the article does not correlate well with its text. In my opinion, there is no proof for using the moment criterion as a predictor of the functionality of the prosthetic foot for transtibial amputees. There is no evidence that a specific parameter of the moment-angle allows patient satisfaction and performance (positively correlated with patient comfort and health) in this article.
Therefore, I recommend changing the title, e.g.: The moment criterion of anthropomorphicity as a quantitative mechanical parameter of prosthetic feet.
Moreover, the presented study has many limitations including a low number of prosthetic feet tested for which the index of Anthropomorphicity has been calculated and the low number of amputees and able-bodied subjects. Additionally, they were examined almost ten years ago (2014). Therefore, if it is possible to extend studies to include data from the literature addressed to modern prostheses, then the authors should do it.
The author should explain in detail how to calculate IA:
- do K1 and K2 represent the slope coefficient of the first and second portion of the regression line) or regression line formula?
- it is best to provide K1 and K2 for the examples presented in Fig. 7 for which the IA indices are calculated.
All figures should be modified and enlarged because they are unreadable! Especially, Figure 7 should be modified and explain the following issues:
- Why is SD for IA only given and not for torque?
- Is the maximum torque selected for one case among all?
- The chart includes five cases, but the authors show/give three. Where does this difference come from and where do the SD readings come from if, for example, FFF is one case? Do SDs refer to three trials?
- Why is the maximum bending moment not normalized?
Finally, are whether the patient's satisfaction and performance examined, the authors claim that there is a correlation between the IA index and the patient's comfort.
The article requires correction according to the above comments before publication.
Reviewer 5 Report
Comments and Suggestions for Authors
biomimetics-2653612-peer-review-v1
The manuscript focus in prosthetics, presenting a distinct viewpoint regarding the application of an anthropomorphicity index and its correlation with prosthetic function and adaptability. The exploration of how a particular aspect, dorsiflexion, relates to overall foot function, especially in the realm of prosthetics, is insightful. Several points need further clarification and restructuring to enhance the comprehensibility and impact of the paper.
Introduction:
1. Integrating sub-headings within the introduction could indeed aid readers in navigating through the different elements being introduced, providing a more structured and digestible framework.
2. Content discussed in Lines 319-328 under the “Discussion” section might be more aptly placed within the introduction, provided it lays foundational context or premises.
Clarity and Terminology
3. Line 355: The term “anthropomorphic characteristics” and its alignment with the "index of anthropomorphicity" require further elucidation. The usage of anthropomorphicity to describe a singular characteristic (dorsiflexion) is seemingly in dissonance with its broad conceptual definition provided by the author. A more thorough explanation or a refinement in terminology might be warranted to prevent potential confusion.
4. Consider providing a clearer delineation between the broader concept of anthropomorphism and its application to the single characteristic of dorsiflexion in the presented index to fortify the argument and ensure clear conveyance to the reader.
Comments on the Quality of English Languageminor editing of English language required